# CRISPR-Cas is beneficial in plasmid competition, but limited by competitor toxin–antitoxin activity when horizontally transferred

David Sünderhauf[1]*, Jahn R. Ringger[1¤], Leighton J. Payne[2], Rafael Pinilla-Redondo[2], William H. Gaze[3], Sam P. Brown[4,5], Stineke van Houte[1]*

1 Environment and Sustainability Institute, University of Exeter, Penryn, United Kingdom, 2 Section of Microbiology, Department of Biology, University of Copenhagen, Copenhagen, Denmark, 3 European Centre for Environment and Human Health, University of Exeter, Penryn, United Kingdom, 4 School of Biological Sciences, Georgia Institute of Technology, Atlanta, Georgia, United States of America, 5 Center for Microbial Dynamics and Infection, Georgia Institute of Technology, Atlanta, Georgia, United States of America

¤ Current address: Department of Environmental Sciences, University of Basel, Basel, Switzerland
* david@sunderhauf.net (DS); C.van-Houte@exeter.ac.uk (SvH)

## Abstract

Bacteria can encode dozens of different immune systems that protect them from infection by mobile genetic elements (MGEs). MGEs themselves may also carry immune systems, such as CRISPR-Cas, to target competitor MGEs. It is unclear when this is favored by natural selection, and whether toxin–antitoxin (TA) systems—common competitive mechanisms carried by plasmids—can alter their efficacy. Here, we develop and test novel theory to analyze the outcome of competition between plasmids when one carries a CRISPR-Cas system that targets the other plasmid. Our mathematical model and experiments using *Escherichia coli* and competing IncP plasmids reveal that plasmid-borne CRISPR-Cas is beneficial to the plasmid carrying it when the plasmid has not recently transferred to a new host. However, CRISPR-Cas is selected against when the plasmid carrying it transfers horizontally, if a resident competitor plasmid encodes a TA system that elicits post-segregational killing. Consistent with a TA barrier to plasmid-borne CRISPR-Cas, a bioinformatic analysis reveals that naturally occurring CRISPR-Cas-bearing plasmids avoid targeting other plasmids with TA systems across bacterial genera. Our work shows how the benefit of plasmid-borne CRISPR-Cas is severely reduced against TA-encoding competitor plasmids, but only when plasmid-borne CRISPR-Cas is horizontally transferred. These findings have key implications for the distribution of prokaryotic defenses and our understanding of their role in competition between MGEs, and the utility of CRISPR-Cas as a tool to remove plasmids from pathogenic bacteria.

**Data availability statement:** The data underlying this manuscript can be found in S1 Data. Code used for data gathering, processing, and visualization can be found in https://doi.org/10.5281/zenodo.18386644.

**Funding:** DS was supported in part by grant MR/N0137941/1 for the GW4 BIOMED MRC DTP, awarded to the Universities of Bath, Bristol, Cardiff and Exeter from the Medical Research Council (MRC)/UKRI. JRR was supported by a summer studentship from the Lister Institute for Preventative Medicine. LJP and RPR were supported by research grant VIL60763 from VILLUM FONDEN. SPB acknowledges funding from the US National Science Foundation (NSF 2321502). WHG and SvH acknowledge funding from the JPI-AMR HARISSA programme (MISTAR; MR/W031191/1), and in addition SvH acknowledges funding from the Biotechnology and Biological Sciences Research Council (BB/R010781/1; BB/S017674/1) and the Lister Institute for Preventative Medicine. The funders had no role in study design, data collection and analysis, decision to publish, or preparation of the manuscript. DS, JRR, and LJP received a salary from JPI-AMR HARISSA, the Lister Institute for Preventative Medicine, and VILLUM FONDEN, respectively. https://www.ukri.org/councils/mrc/; https://lister-institute.org.uk/; https://villumfonden.dk/en; https://www.nsf.gov/; https://jpiamr.eu/; https://www.ukri.org/councils/bbsrc/.

**Competing interests:** I have read the journal's policy and the authors of this manuscript have the following competing interests: SPB is a member of PLOS Biology's Editorial Board. The other authors declare that no competing interests exist.

**Abbreviations:** MGEs, mobile genetic elements; PCN, plasmid copy number; PSK, post-segregational killing; TA, toxin–antitoxin.

## Introduction

Mobile genetic elements (MGEs) are widely distributed throughout the prokaryotic tree of life and shape the ecology and evolution of bacterial populations. Conjugative plasmids are the MGEs most proficient at horizontally transferring genes (including antimicrobial resistance genes) through bacterial communities [1], and frequently co-infect the same bacterial cell [2]. Prokaryotic immune systems are abundant in prokaryotic genomes and can alter not only how the bacterial host interacts with MGEs, but also how co-infecting MGEs interact with each other [3,4]. However, the unique selection pressures faced by immune systems that are encoded on MGEs have not been investigated.

CRISPR-Cas is an adaptive immune response whereby a ribonucleoprotein complex acts as an RNA-guided DNA or RNA nuclease and specifically recognizes and cleaves an invading MGE (and/or its transcripts), based on a short sequence stored in the CRISPR locus as a "spacer" during a previous infection (reviewed in [5]). CRISPR-Cas systems are found on ~3% of plasmids [6], where they play a role in competition between plasmids (reviewed in [7]): plasmids' CRISPR spacers tend to match other competing plasmids [6,8].

In response, plasmids may invest in traits that give them a fitness advantage. These can be either through direct interference with CRISPR-Cas, such as anti-CRISPRs [9], or through CRISPR-Cas-independent features such as plasmid incompatibility [10], Bet/Exo systems [11], entry exclusion [12], and toxin–antitoxin (TA) systems [13]. Of these, TA systems are highly prevalent and carried by ~35% of plasmids [14]. TA systems function by simultaneous production of a toxin and a short-lived antitoxin. If the plasmid—and therefore the TA genes it carries—are removed, the toxin will be freed up to harm or kill the bacterial host. This addictive nature of TA systems means their removal is strongly selected against [13] (including removal by chromosomally-encoded CRISPR-Cas [15,16]), and this phenomenon is theorized to be beneficial to plasmid competition [17–19]. In line with this, TA systems were identified as a barrier to plasmid removal when using CRISPR-Cas or incompatibility-based technologies [20,21].

Both TA systems and CRISPR-Cas immune systems are important during within-cell plasmid competition [8,19], but it is unclear how selection acts on these traits when they directly compete with one another. To test this, we constructed a mathematical model to predict the outcome of plasmid competition between a CRISPR-Cas-bearing and a TA-bearing plasmid within an individual host, dependent on whether the CRISPR-Cas plasmid was acting offensively (i.e., invaded a new host) or defensively (i.e., protected its host from competitor invasion). We tested model predictions using an experimental system where a CRISPR-Cas-bearing and a TA-bearing IncP plasmid competed for *Escherichia coli* hosts in similar ecological scenarios. Finally, we further tested our theory and generalized our experimental results by bioinformatically analyzing TA carriage of target plasmids of naturally occurring plasmid-borne CRISPR-Cas systems. Together, our theory, experiments, and bioinformatics analyses show that when CRISPR-Cas-bearing plasmids are acting offensively and invading a new host, the otherwise universal benefit of

CRISPR-Cas to plasmid competition is limited by TA carried on competitor plasmids. This work highlights the limitations of CRISPR-Cas in plasmid competition and underlines the advantages of TA systems to plasmid spread and persistence.

## Results

### Mathematical modeling predicts that the benefit of CRISPR-Cas depends on toxin–antitoxin (TA) activity of competitor plasmids

Both CRISPR-Cas and TA systems are implicated in plasmid-level competition [8,19]. In a mathematical analysis, we set out to formalize the relative benefits of CRISPR-Cas to a plasmid when competing with a plasmid encoding TA, depending on asymmetries due to distinct offensive versus defensive roles of CRISPR-Cas and TA. A defensive role refers to a plasmid remaining resident in a cell line. In contrast, an offensive role refers to horizontal transfer of a plasmid to a new host and becoming a newly arrived invader in a cell.

To formalize plasmid competition dynamics, we begin with a simple conceptualization of plasmid competition (Fig 1A and 1B). Our approach focuses on cells co-infected with both plasmids ($CT$ and $TC$ cells), with plasmids differentiated by their order of arrival. $C$ represents cells where a CRISPR-Cas plasmid is resident, while $T$ represents cells where a TA plasmid is resident. In the event $C$ cells are infected by a TA-plasmid, we have a $CT$ cell containing both a resident CRISPR-Cas plasmid and an invading TA plasmid. Conversely, $TC$ cells are created when a $T$ cell acquires a CRISPR-Cas plasmid.

Our model defines transition rates from co-infected states ($CT$ or $TC$) to singly infected states ($C$ or $T$) (arrows in Fig 1A and 1B). These transition rates are defined by plasmid segregational loss (baseline rate $s$), modified by $C$- and $T$-specific traits of CRISPR-Cas-induced enhanced segregation $x$ (blue in Fig 1A; $x > 1$ implies multiplicatively increased segregation rate $x \cdot s$) and TA-induced probability of post-segregational killing (PSK) $y$ (red in Fig 1A). Asymmetries in parameter values between resident and invader roles are captured by subscripts $r$ and $i$. In the event of PSK, liberated resources (nutrients, space) can enhance the growth of remaining $C$ or $T$ cells, weighted by parameter $f$ [19] where $f_r$ refers to the benefit given to cells containing the previously resident plasmid ($T$ for $TC$ cells and vice versa). Given lower plasmid copy number and lower gene expression for invading versus resident plasmids [22], our default assumptions are $s_i > s_r$; $x_r > x_i \geq 1$; $1 > y_r > y_i \geq 0$. Given spatial structuring so that $T$ cells are enriched in the neighborhood of $TC$ cells (and vice versa for $C$ near $CT$ cells), we further assume $1 > f_r + f_i > f_r > f_i \geq 0$. The resulting transition rates (Fig 1A and 1B, equations E1–E4) are used to define CRISPR-Cas competition summary metrics $\Delta_C$, which capture whether CRISPR-Cas investment $x$ results in more transitions to $C$ than to $T$ cells ($\Delta_C > 0$), or vice versa ($\Delta_C < 0$). These metrics are calculated for both the context when the CRISPR-Cas plasmid is resident (Fig 1A–1C, equation E5), and when it is an invader (Fig 1B–1D, equation E7).

Analysis of $\Delta_C$ of CRISPR-Cas plasmids provided the following core predictions concerning the impact of CRISPR-Cas and TA investments ($x$ and $y$) on their relative competitive success (derivations in S2 Text): Firstly, in the absence of PSK (no TA; $y = 0$), increasing investment in CRISPR-Cas $x$ always increases CRISPR-Cas competitive success, regardless of whether in a defensive (Fig 1C) or offensive role (Fig 1D). Secondly, in the presence of PSK ($y > 0$), increasing CRISPR-Cas investments is still beneficial in a resident context (Fig 1C), but can become detrimental in an offensive role, given sufficiently strong resident TA $y_r$ (Fig 1D, area above the dashed line). Analytical conditions for this qualitative shift in response are defined in S2 Text.

In conclusion, our model predicted an asymmetric outcome of plasmid competition, depending on CRISPR-Cas mode of action. When resident, investing in stronger CRISPR-Cas always benefitted its own plasmid. In offense CRISPR-Cas switched from an asset to a liability in plasmid competition, dependent on a sufficiently strong TA response on the competitor plasmid.

### Experimentally competing a CRISPR-Cas and TA plasmid showed a universal CRISPR-Cas benefit in defence, but not in offense

Our mathematical modeling predicts that the benefits of a plasmid-borne CRISPR-Cas system are limited during competition with TA plasmids. We sought to test these predictions experimentally in a model system where we could modulate

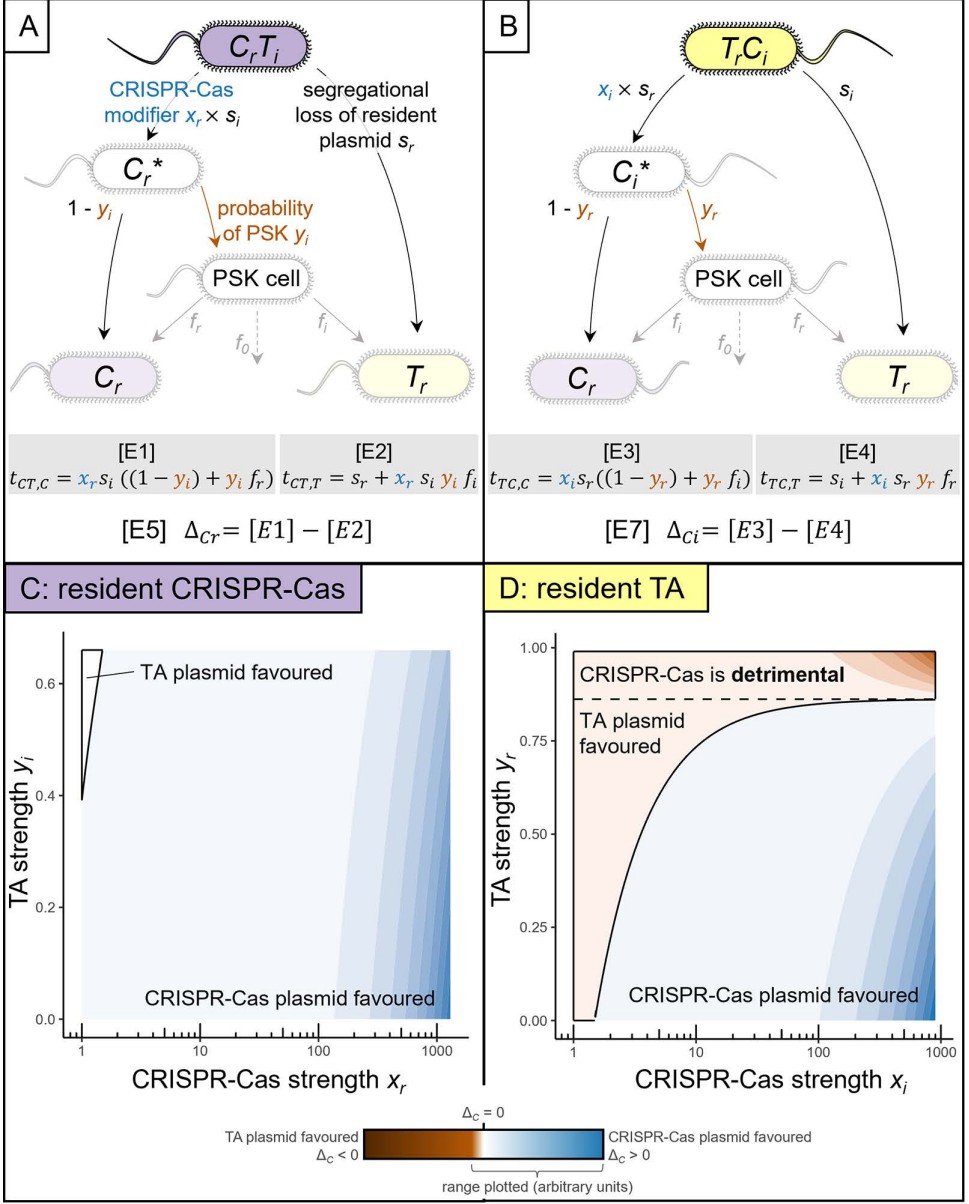

**Fig 1. Mathematical modeling predicts that CRISPR-Cas has a greater potential to benefit its plasmid in defence than in offense. (A,B)** Structure of the mathematical model when a CRISPR-Cas plasmid is the resident (A) or the invader plasmid (B). We consider seven distinct cell types: $C$ (CRISPR-Cas bearing) and $T$ (TA-bearing) plasmids reside in host cells; $CT$ denotes a $C$ cell that has been invaded by a $T$ plasmid and vice versa; $C_r^*$ and $C_i^*$ cells have undergone segregational loss of a $T$ plasmid, and the fate of PSK is not yet resolved; PSK cells are dead cells after post-segregational killing. When the plasmids co-reside, they are subject to basic biological parameter $s$ (segregational loss). CRISPR-Cas and TA alter parameter $s$ by their own modes of action $x$ and $y$, respectively. Parameter $f$ describes the positive effect PSK cells have on growth of nearby cells, where $f_r$ describes the benefit to cells containing the previously resident plasmid. All parameters adopt distinct values when their respective plasmids are resident ($r$) or invasive ($i$). **(C, D)** Plasmid competition summary metric of the $C$ plasmid, in either resident ($\Delta_{Cr}$) or invader role ($\Delta_{Ci}$). Values >1 describe plasmid competition resolved in favor of the CRISPR-Cas plasmid, and values <1 describe plasmid competition resolved in favor of the TA plasmid (highlighted area) as a function of strength of CRISPR-Cas system $x$ and strength of TA system $y$. Both $x$ and $y$ adopt different maximum values depending on which plasmid is resident (maximum $y_i$ = 0.66; maximum $y_r$ = 0.99; maximum $x_r$ = 1,300; maximum $x_i$ = 900). Parameter choices are specified in Methods.

CRISPR-Cas and TA strength in a binary way. To this end, we used competitor plasmids pKJK5::csg (IncP1ε) and RP4 (IncP1α) in *E. coli* hosts as models to examine how selection acts on plasmid-encoded CRISPR-Cas under defensive and offensive modes of action (Fig 2A and 2B). Due to plasmid incompatibility resulting in segregational loss, these plasmids cannot be co-maintained in the same host indefinitely but can co-exist for several generations [23], giving an ample window of opportunity for plasmid competition within the same bacterial host. Plasmid pKJK5::csg was engineered to carry a minimal Type II CRISPR-Cas9 system [24], and we turned CRISPR-Cas activity on or off by using two isogenic plasmid variants that only differed in their respective fixed targeting specificity (Fig 2A, see Methods). Competitor plasmid RP4 naturally carries the Type II TA system *parABCDE*, where *parABC* form a segregational stability system and *parDE* code for a TA system that inhibits DNA gyrase, leading to cell filamentation and cell death of plasmid-free segregants [25–27].

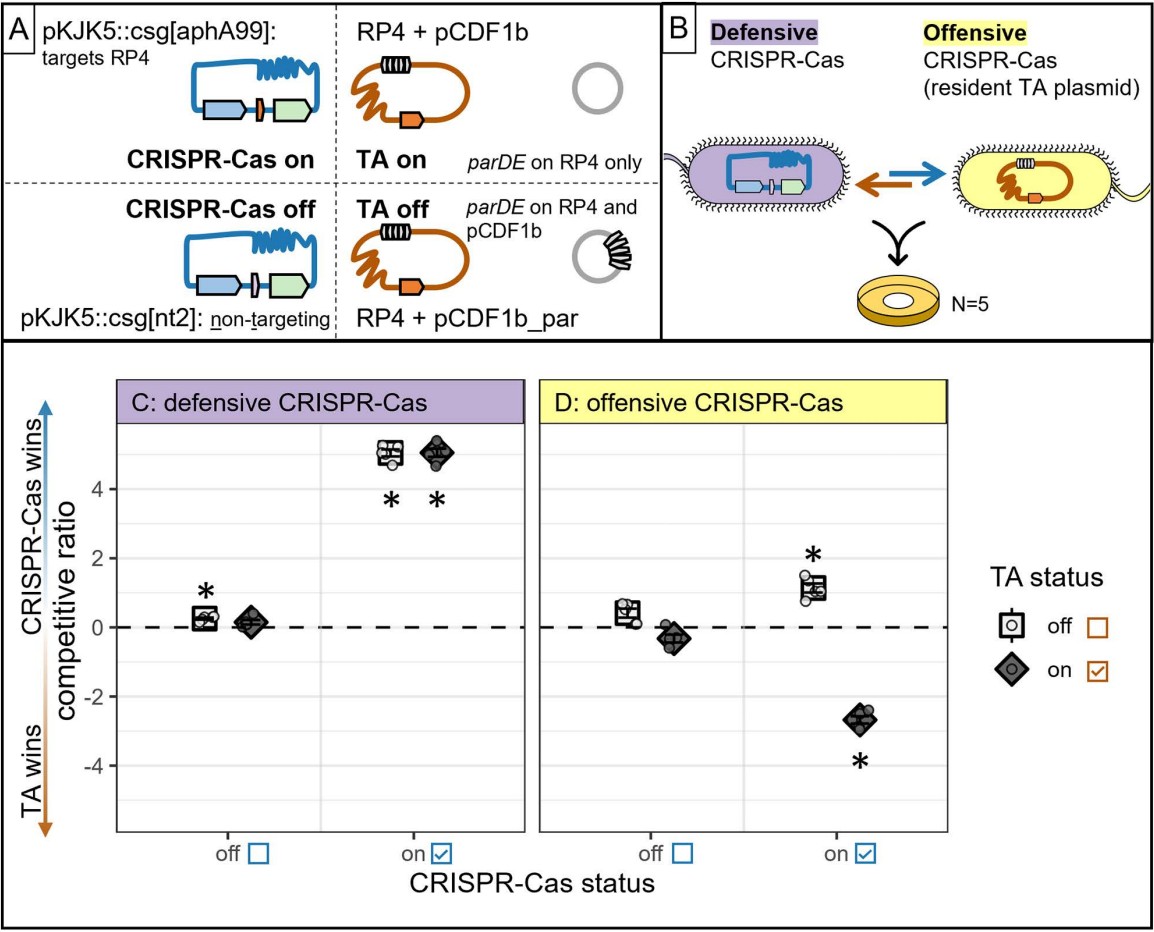

**Fig 2. CRISPR-Cas is beneficial in defence but limited by TA in offense in experimental plasmid competition. (A)** CRISPR-Cas and TA activity were tweaked on competitor plasmids in a binary manner by switching each system on or off. pKJK5::csg carries a gene cassette encompassing *cas9*, *sgRNA, GFP* with either an RP4-targeting or non-targeting guide, RP4 naturally carries TA system *parABCDE*. **(B)** Plasmids were competed using two *Escherichia coli* DH5α hosts with different plasmid content, allowing us to assess competitive outcome under defensive and offensive CRISPR-Cas action. After filter mating, plasmid content of each host was assessed by selective plating. **(C, D)** Mean±standard error of the competitive ratio (log odds ratio of pKJK5-carrying hosts/RP4-carrying hosts) describes outcome of plasmid competition ($N=5$). Values >0 indicate CRISPR-Cas plasmid pKJK5 winning the competition, and values <0 indicate TA plasmid RP4 winning the competition for a certain host. The dashed line indicates a neutral outcome with competitive ratio=0. Data are presented for treatments in which CRISPR-Cas and TA activity were toggled on or off in all combinations. Stars indicate significant differences from 0 as assessed by *T* test and Bonferroni adjustment for multiple testing; *$p<0.0063$ with $α=0.0063$; see Table 3 for all *p* values. The data underlying this Figure can be found in S1 Data.

We turned RP4's TA activity off by supplying a bystander expression vector engineered to carry RP4's *par* TA operon (pCDF1b_par) to all potential plasmid hosts, thus allowing supply of the antitoxin *in trans* and negating the PSK effect of RP4's TA system. As a control, empty vector pCDF1b was supplied in all treatments where TA remained turned on (Fig 2A). These vectors were maintained throughout all experiments in the absence of antibiotic selection (S1 Fig).

To assess plasmid competition outcome under different modes of CRISPR-Cas action, we set up mating assays including two differentially tagged *E. coli* DH5α hosts with different plasmid content (Fig 2B). In this way, we could track plasmid competition when CRISPR-Cas was acting defensively (a pKJK5 host was invaded by RP4) and when CRISPR-Cas was acting offensively (an RP4 host was invaded by pKJK5). After solid-surface filter mating, we calculated the outcome of plasmid competition by measuring the relative abundance of each plasmid in each host by selective plating (competitive ratio; the log odds ratio of the proportion of pKJK5 over RP4 carriage within hosts; positive values indicate that CRISPR-Cas plasmid pKJK5 was more abundant in its host, and negative values indicate that TA plasmid RP4 was more abundant).

This analysis revealed that when CRISPR-Cas was acting defensively, the CRISPR-Cas plasmid pKJK5 won the competition in every case where CRISPR-Cas was switched on (Fig 2C; competitive ratio $= 5.04 \pm 0.10$ and $5.05 \pm 0.12$, respectively; both significantly higher than 0 with $p < 0.001$ after $T$ test and Bonferroni adjustment for multiple testing with $\alpha = 0.0063$). In stark contrast, when CRISPR-Cas was acting offensively, switching on CRISPR-Cas had a differential effect depending on TA status of the competitor plasmid. When TA was absent from the competitor, switching on CRISPR-Cas brought a benefit to pKJK5 (Fig 2D; competitive ratio $= 1.14 \pm 0.13$; $p < 0.001$). However, when the competitor plasmid's TA was switched on, switching on CRISPR-Cas was clearly detrimental to pKJK5 (competitive ratio $= -2.68 \pm 0.10$; $p < 0.001$). In line with these observations, statistical modeling confirmed that TA status was not a significant determinant of plasmid competition outcome in defence ($p = 0.40$ modeling coefficient of GLM; see Methods for details), but only in offence ($p = 0.00053$). CRISPR-Cas status contributed significantly to plasmid competition outcome in both plasmid hosts ($p < 2 \times 10^{-16}$ in defence; $p = 0.0059$ in offence).

Next, we tested whether these results were upheld in a more complex model system containing a third mating partner, which was initially plasmid-free (S2 Fig). In this expanded model system, plasmid behavior within the defensive and the offensive CRISPR-Cas host followed the established behavior and CRISPR-Cas was detrimental when competing with a TA-on plasmid in offence, but not in defence (S2C and S2D Fig; S1 Text). In the third, initially plasmid-free host, CRISPR-Cas could act defensively and offensively. Switching on CRISPR-Cas on its own benefitted pKJK5 (S2E Fig; competitive ratio $0.73 \pm 0.1$; $p = 0.0019$). When TA was switched on alongside this, there was no clear winner of the competition (competitive ratio $= -0.21 \pm 0.096$; $p = 0.1$), however, CRISPR-Cas still provided a benefit to pKJK5 relative to the scenario where CRISPR-Cas was switched off, where TA plasmid RP4 won the competition (competitive ratio $= -0.94 \pm 0.066$; $p < 0.001$).

In addition, to determine whether our observed effects are driven by RP4's *parABCDE* TA system rather than other RP4-encoded mechanisms such as entry exclusion or incompatibility, we verified our conclusions by cloning the *par* operon onto a series of cloning vectors with different backbones of varying incompatibility group and offensively competing pKJK5 against these (S3 Fig, S1 Text). This confirmed that CRISPR-Cas on its own benefits pKJK5, but if a competitor plasmid carries the *par* TA system, CRISPR-Cas is detrimental to pKJK5 in offense—regardless of competitor plasmid backbone.

Together, our experiments show that a defensive CRISPR-Cas system was always beneficial to its host plasmid—regardless of competitor TA status—in agreement with our core modeling predictions. These results were upheld during a mixed mode of CRISPR-Cas action, where CRISPR-Cas was never detrimental. In contrast, we found that the benefit of an offensive CRISPR-Cas system to its host plasmid depended on payload genes encoded on competitor plasmids; namely, a TA system on a competitor means that offensive CRISPR-Cas was detrimental to its plasmid.

## Plasmid-borne CRISPR-Cas systems preferentially target plasmids lacking TA systems

Our model and experiments suggest that the offensive benefits of plasmid-borne CRISPR-Cas systems are limited to conflicts where the competitor plasmid does not encode a TA system. To test whether this constraint is reflected in natural systems, we analyzed genomic data to assess the association between CRISPR-Cas targeting and TA presence. If the

fitness benefit of CRISPR-Cas systems in plasmid competition is limited by TA systems, we hypothesized that naturally occurring plasmid-borne CRISPR-Cas systems would preferentially target plasmids without TA systems.

To answer this question, we analyzed existing data on the plasmid-targeting spacers of plasmid-borne CRISPR-Cas systems [6], focusing on identifying which of these targeted plasmids encoded TA systems. The resulting targeting network provided a comprehensive picture of plasmids that were targeted by plasmid-borne CRISPR-Cas systems ($N=1,340$), and the TA systems these targeted plasmids encoded (Fig 3A). Type II TA, including *parDE* ($N=565$), were most commonly identified. Within this network, 34% of plasmid-borne CRISPR-Cas systems were Type IV systems (S5A Fig), which function by repressing target gene expression, rather than causing nucleolytic cleavage [8,28]. This mode of action may allow targeted plasmids to persist without triggering PSK, potentially obscuring the relationship between targeting and TA presence. Indeed, Type IV-A1 and IV-A3-targeted plasmids have been shown to persist when targeted in non-essential regions [8,29]. Therefore, we repeated the analysis by partitioning the dataset into Type IV and non-Type IV systems. Restricting the data to non-Type IV systems reduced the number of targeted plasmids to 463, of which only 11% ($N=51$) encoded TA systems, and none encoded *parDE* (S5B Fig).

Correlational analyses on the partitioned data (all CRISPR-Cas types, Type IV only, and non-Type IV) yielded contrasting results. In all cases, Pearson's $\chi^2$ tests confirmed a significant relationship between CRISPR-Cas targeting and TA presence ($\chi^2=94.08$, 334.48, and 171.83, respectively; $p<2.2\times10^{-16}$), with small effect sizes ($\varphi=0.073$, 0.137, and 0.098, respectively). When considering all plasmid-borne CRISPR-Cas types or Type IV only, TA-encoding plasmids were targeted more often than expected under the null hypothesis of no relationship between CRISPR-Cas targeting and TA carriage (residuals $=7.21$ and 13.82; $p=5.71\times10^{-13}$ and $1.04\times10^{-43}$, respectively; Fig 3B). This contradicts our hypothesis that plasmid-borne CRISPR-Cas systems should preferentially target plasmids lacking TA systems. In contrast, when we excluded Type IV CRISPR-Cas systems, the association between CRISPR-Cas targeting and TA carriage reversed: fewer TA-encoding plasmids were targeted than expected (residual $=-10$; $p=1.49\times10^{-23}$), consistent with our experimental data and model predictions.

Having established that non-Type IV plasmid-borne CRISPR-Cas systems preferentially target plasmids lacking TA systems, we next considered the few cases where they do target TA-encoding plasmids. Based on the expectation that such encounters would trigger PSK, we hypothesized that non-Type IV CRISPR-Cas plasmids would be more likely than Type IV CRISPR-Cas plasmids to encode compatible antitoxin families when targeting TA-encoding plasmids. To test this, we compared the frequency of compatible antitoxin families carried by plasmids with non-Type IV versus Type IV CRISPR-Cas systems when targeting TA+ plasmids. Compatible antitoxin families were present in 22% (15/69) of cases of non-Type IV targeting, but only 6% (352/5914) of Type IV cases (Fig 3C). Fisher's exact test confirmed enrichment in nucleolytic systems (OR = 4.4, 95% CI $=2.3–8$, $p=1.3\times10^{-5}$). A complementary binomial test, using the Type IV frequency (6%) as the empirical null, also showed significant enrichment of compatible antitoxin families in cases of non-Type IV targeting (observed $=22\%$, 95% CI $=13\%–33\%$, $p=1.1\times10^{-5}$).

Overall, these results indicate that nucleolytic plasmid-borne CRISPR-Cas systems preferentially target plasmids lacking TA systems. This was observed with a modest effect size, which may be due to an additional propensity of plasmids with nucleolytic CRISPR-Cas systems to carry their own TA systems of the same family as found on target plasmids, hypothetically ameliorating TA activity. These findings support our model and extend our experimental findings to natural systems across diverse CRISPR-Cas and TA system types. We conclude that nucleolytic, plasmid-borne CRISPR-Cas systems provide significant defensive benefits, even in the face of TA. In contrast, their benefits in offence are constrained by the presence of TA systems on competitor plasmids.

## Discussion

Here, we examined the limitations of CRISPR-Cas in plasmid competition. Mathematical modeling predicted that when plasmids compete within co-infected cells, CRISPR-Cas can provide a benefit to the plasmid it resides on. However, when

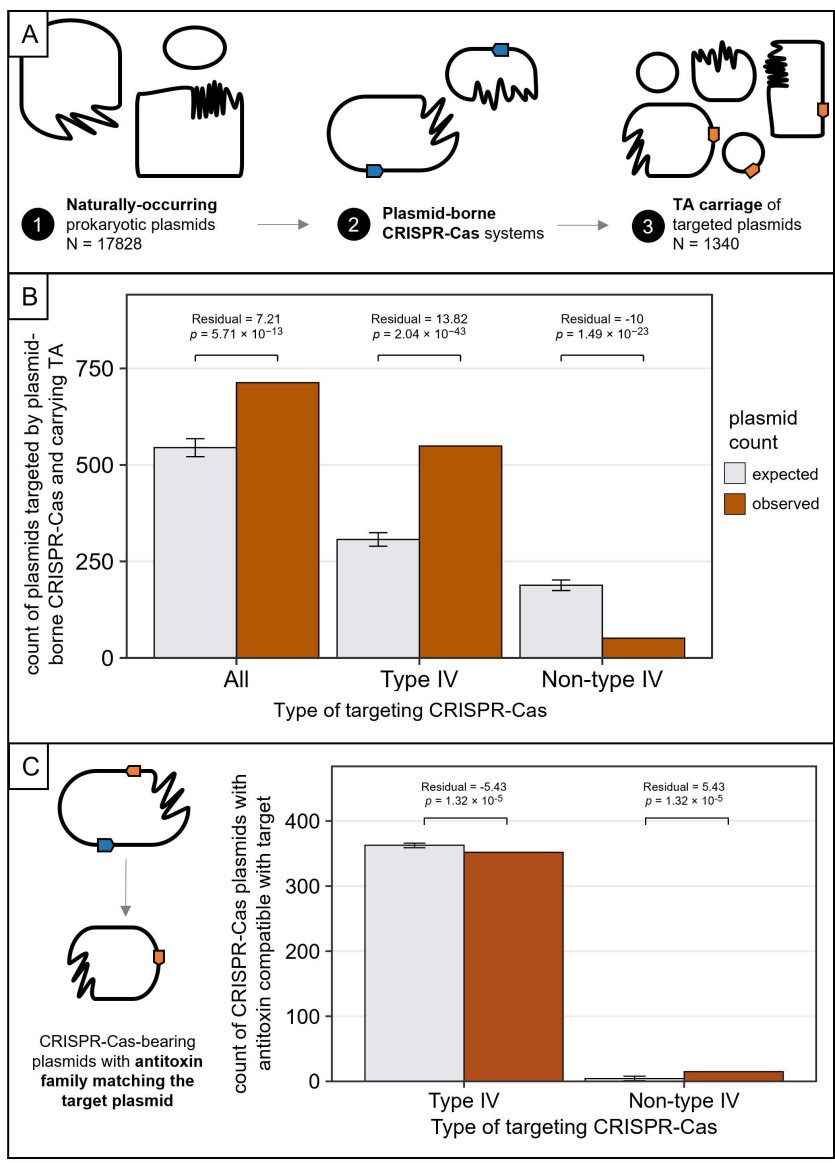

**Fig 3. Plasmid-borne CRISPR-Cas systems (excluding Type IV) avoid targeting plasmids with toxin–antitoxin (TA) systems. (A)** Naturally-occurring plasmid-borne CRISPR-Cas systems and the plasmids they target [6] were analyzed for TA carriage. **(B)** Comparison of observed vs. expected counts of targeted plasmids encoding TA by CRISPR-Cas type. Brown bars show the observed counts of plasmids in each group compared to the expected counts (gray bars) under the null hypothesis of independence between targeting and TA presence. Error bars represent standard error. Residuals and $p$ values are presented after a Pearson's $\chi^2$ test. Full contingency tables for all categories of plasmids can be found in S6 Fig (targeted or not targeted by CRISPR-Cas, coding or not coding for TA). **(C)** Comparison of observed vs. expected counts of CRISPR-Cas bearing plasmids carrying antitoxins of the same family as the target plasmid's TA system. Brown bars show the observed counts of plasmids in each group compared to the expected counts (gray bars) under the null hypothesis of independence between CRISPR-Cas targeting and the presence of compatible antitoxin families on the CRISPR-Cas plasmid. Error bars represent the 95% interval of counts expected under the null hypothesis of independence, calculated from the hypergeometric distribution. Residuals and $p$ values are presented after a Fisher's exact test. The data underlying this Figure can be found in S1 Data.

the competitor plasmid encodes a strong TA system, CRISPR-Cas is only beneficial when defensive and resident in a host. Our experimental model system showed the same outcome as these core modeling predictions: offensive CRISPR-Cas becomes a liability when targeting a resident TA plasmid, while defensive and co-invading CRISPR-Cas provides a benefit even when competing against a TA plasmid. Finally, we further tested our theory and generalized our experimental findings by bioinformatically showing that plasmid-borne nucleolytic CRISPR-Cas systems avoid targeting other plasmids carrying TA systems or tend to carry their own TA systems of compatible families. In line with our experimental data, recent single-cell observations showed that chromosomal CRISPR-Cas systems (i.e., lacking an offensive capability) protected a population from invasion by TA-carrying plasmid RP4, even when TA was expressed [16].

TA was previously shown to be crucial to plasmid establishment, spread, and maintenance [17–19]. Our data adds to this broad utility of TA throughout a plasmid's life cycle, as we found TA contributing to its plasmid competitiveness in nearly all experimental scenarios (Figs 2 and S2). Underlining this, our model predicted that, unlike CRISPR-Cas (the benefit of which diminished when competing TA was stronger), TA became more beneficial in the presence of a stronger CRISPR-Cas response on the competitor plasmid (Fig 1; S2 Text). Regardless, TA and CRISPR-Cas show striking similarity: they benefit their own plasmid more when acting defensively than offensively (Figs 1 and 2). It is well established that plasmids are more fit when residing in a host cell than when conjugating into a new host [30]. Conjugation itself is costly (e.g., by SOS-response activation [31]), a novel plasmid is associated with a high transient fitness cost [32], and existing plasmid-host pairs can benefit from compensatory mutations to ameliorate fitness costs [33]. Here, we add to this body of evidence by showing that specific plasmid-encoded genes, namely CRISPR-Cas and TA, also provide a higher benefit to vertically transferred plasmids than to those that are horizontally transferred when facing plasmid competition.

Recent modeling showed that plasmid cost is a key determinant of TA plasmid establishment in a bacterial population carrying (chromosomally encoded) CRISPR-Cas systems [34]. In support of this, the cost of plasmid co-infection was identified as a hurdle to plasmid establishment [35]. We do not directly measure plasmid costs in our experimental system, but our model assumes the presence of both CRISPR-Cas and TA is costly due to PSK. Experimentally, we are likely overinflating TA strength compared to natural populations: DH5α, our experimental host, is a recA− E. coli strain. RecA is implicated in parDE TA function, where recA− strains typically see survival rates ~3 orders of magnitude lower than recA+ strains [26]. Nevertheless, our bioinformatic analysis confirms the generalisability of our experimental results by revealing a negative association between plasmid-borne CRISPR-Cas targeting and competitor plasmid TA carriage.

Our findings help explain the distribution and mobility of microbial immune systems, where certain systems are more commonly found on specific MGEs rather than the bacterial chromosome [36,37]. Due to their offensive capability, plasmid-borne CRISPR-Cas systems are under different selective pressures than chromosomally encoded CRISPR-Cas systems—and therefore it is unsurprising that certain types of CRISPR-Cas are preferentially found on plasmids [6]. In our bioinformatics analysis, only nucleolytic (non-Type IV) plasmid-borne CRISPR-Cas systems avoid targeting plasmids carrying TA systems. Based on this, we hypothesize that plasmid-borne CRISPR-Cas systems can adopt different evolutionary strategies in light of competitor TA: a plasmid-borne CRISPR-Cas system could avoid targeting TA plasmids or become associated with compatible TA systems to overcome limitations caused by TA in offense, as showcased in our data and bioinformatics analysis. An alternative evolutionary scenario could involve the loss of nucleolytic activities, as seen for plasmid-borne Type IV and Type V-M [38] CRISPR-Cas systems. These would not have limitations in offense and can target TA-carrying plasmids without paying the costs of PSK induction. In addition, Type IV CRISPR-Cas systems may overcome TA activity by co-opting their host's spacer acquisition machinery [8,39]—this may strengthen Type IV CRISPR-Cas's role in defence (while resident in its host, the CRISPR-Cas system can acquire new spacers to defend from MGEs), while dampening its role in offense (when moving to a new host, the CRISPR-Cas system cannot acquire new spacers until it is expressed and therefore cannot target a newly encountered plasmid already present in the new host). The proposed evolutionary history of Type IV CRISPR-Cas systems supports the hypothesis that plasmid-borne CRISPR-Cas systems adopt different evolutionary strategies to remain effective in light of TA: nucleolytic Type III

CRISPR-Cas lost its HD nuclease domain and diverged towards gene silencing activity by association with helicase DinG over evolutionary timescales [40]. Overall, the association between TA and the effectiveness of CRISPR-Cas in plasmid competition helps explain the distribution and mobility of bacterial defence systems and the propensity of certain defences to associate with particular MGEs.

Throughout this work, we assayed plasmid competition outcome within individual plasmid hosts. Despite this limitation, our work could inform predictions of community-level plasmid persistence. On a population level, within-host plasmid competition may be particularly relevant to highly permissive plasmid hosts—bacteria in which the cost of plasmid maintenance is low and which can cause persistence of plasmids on a community level [41,42], or in hosts closely associated with certain plasmid lineages, such as clinically problematic bacterial strains of *E. coli* Sequence Type 131 and their IncF plasmids [43]. A resident CRISPR-Cas plasmid in such highly permissive plasmid hosts may prevent their colonization by TA-encoding competitor plasmids and thus prevent TA plasmid dissemination throughout the microbial community. In addition to making predictions about plasmid prevalence, our work can inform the use of CRISPR-Cas delivery tools (engineered plasmids used to directly kill or resensitise bacteria to antibiotics through removal of antimicrobial resistance-encoding plasmids [24,44]). Our work indicates that CRISPR-Cas delivery tools will be more effective when delivered prophylactically, i.e., when CRISPR-Cas is allowed to act defensively rather than offensively. When they are required to act offensively, other factors such as target plasmid copy number and compatibility can inform optimal CRISPR-Cas delivery tool design [45].

Here, we described how the relevance of plasmid-borne CRISPR-Cas to plasmid competition depends on its mode of action: defensive CRISPR-Cas is universally beneficial, but the benefit of offensive CRISPR-Cas can be limited by TA carried on competitor plasmids. This work underlines the utility of TA systems to plasmid competition and explains their abundance, and may help predict which plasmids can become established in clinically relevant species and communities. Further, we help explain the distribution of bacterial immune systems by uncovering under which conditions plasmid-borne CRISPR-Cas systems are selected for, which will also inform the use of CRISPR-Cas-based technologies.

## Methods

### Mathematical model

Mathematical modeling was carried out using Wolfram Mathematica v14.1. Equations E5 and E7 (see Fig 1; S2 Text) were imported into R version 4.3.2. Output plots were generated using RStudio version 2023.09.1 + 494 with the following packages: dplyr, ggplot2, and ggnewscale.

In order to visualize our mathematical results in Fig 1, we set parameters as detailed in Table 1.

### Plasmids and strains

Plasmids and strains used in this study are summarized in Table 2. *E. coli* DH5α strains DH5α::CmR, DH5α::GmR, and DH5α::SmR were constructed by electroporating DH5α with pBAM1-Gm, pBAM1-Sm, and pBAM1-Cm [48,49], respectively, and selecting for resistant recombinants; resistance gene insertion was confirmed by PCR and by verifying their respective resistant phenotypes after transfer in the absence of antibiotics.

RP4-targeting pKJK5::csg[aphA99] ('CRISPR-Cas on' version of pKJK5) was constructed using λ-red mediated recombineering and pKJK5 as a template, as described in [24] with the exception that *cas9*, *sgRNA* (programmed to target *aphA* with specificity AAATGGGCGCGTGATAAGGT) and *gfp* were inserted into pKJK5's *intI1* gene. Non-targeting control pKJK5::csg[nt2] ('CRISPR-Cas off' version of pKJK5) was previously published as pKJK5::*gfp*^PL [50] and carries a random non-functional *sgRNA*.

Vector pOGG99_par was constructed by deleting genes from pOPS0378 [51] to generate a minimal vector: *mCherry* was removed by digestion with EcoRI, extraction, and re-ligation of the 6,043 bp band. Next, site-directed mutagenesis of *aphA*

**Table 1. Default parameter values and rationales for graphical analyses.**

| Baseline segregation rate $s$; $s_i > s_r$ | |
| --- | --- |
| $s_r = 1$<br>$s_i = 1.49$ | Segregation rate when resident is set at $s_r = 1$ as a baseline. In comparison, $s_i$ adopts a higher value, as newly invaded single-copy plasmids are more likely to be lost by segregation than resident multi-copy plasmids. Large plasmids, which will be more likely to carry systems such as CRISPR-Cas and TA, are most frequently found at an average plasmid copy number (PCN) of 1.49 [46]. Note that setting $s_i = 1.49$ is a conservative estimate, as gene expression (e.g., of stable segregation systems) together with copy number of newly invaded plasmids is lower compared to established plasmids [22]. |
| **PSK benefit to nearby kin cells $f$; $1 > f_r + f_i > f_r > f_i \geq 0$** | |
| $f_r = 0.2$<br>$f_i = 0.04$ | As a baseline, $f_r$ captures the benefit resources freed up by post-segregational killing bring to neighboring cells. $f_r = 0.2$ assumes that PSK resources contribute to 20% as much growth of single-infected cells as loss of the competitor plasmid by segregation. The difference between $f_r$ and $f_i$ captures how well-structured an environment is, with $f_i = 1/5 \times f_r$, we are assuming a dead cell is five times more likely to be surrounded by kin- than non-kin cells. A spatially structured environment is crucial to TA competitiveness [19]. |
| **Strength of CRISPR-Cas $x$; $x_r > x_i \geq 1$** | |
| $x_r = 1{,}300$<br>$x_i = 900$<br>(max. values) | $x$ describes the multiplicative effect CRISPR-Cas has on its competitor plasmid's segregation rate (by cleaving it). In an analysis of plasmids targeted by a chromosomal CRISPR-Cas system, PCN of different plasmids was reduced to ~0.077%–0.06% by CRISPR-Cas action depending on plasmid [47]. Thus, we estimate that CRISPR-Cas activity accelerates competitor segregation 1300-fold ($\approx 1/0.00077$). CRISPR-Cas activity on a newly invaded plasmid $x_i$ is scaled by PCN, leading to $x_i = 1/1.49 * x_r \approx 900$. |
| **Strength of TA $y$; $1 > y_r > y_i \geq 0$** | |
| $y_r = 0.99$<br>$y_i = 0.66$<br>(max. values) | We set the maximum $y_r$ value based on an average survival rate of post-segregational killing of 7 different TA systems of ~0.0067 [26]. We scaled $y_i$ by PCN, leading to $y_i = 1/1.49 * y_r \approx 0.66$. |

was carried out using primers pOGG0_mut_fw and pOGG0_mut_rv and Thermo Scientific's site-directed mutagenesis kit according to manufacturer's instructions. In this way, the nucleotide at position 96 within *aphA* was silently mutated from C to G to bring the gene sequence in line with that found on our laboratory's version of RP4 and allow targeting by pKJK5::csg[a-phA99]. Successful mutation was confirmed by Sanger sequencing of the finished plasmid using primer pOGG0_sequence. pOGG99 was constructed by excising *parABCDE* from pOGG99_par: pOGG99_par was digested with BlpI and BglII, the 3,664 bp fragment was extracted and religated with annealed and phosphorylated linker oligos (pOGG99_parRemoval_top & btm). This yielded pOGG99, a version of pOGG99_par where *parABCDE* are replaced with a multiple cloning site module.

Vector pHERD99 was constructed by annealing oligos aphA99PAM_top and aphA99_btm and inserting them into pHERD30T's HindIII and KpnI restriction sites following standard molecular cloning protocols. Vector pSEVA251-99 was constructed by site-directed mutagenesis of pSEVA251's *aphA* gene as described for pOGG99_par above.

Vectors pCDF1b_par, pHERD99_par, and pSEVA251–99_par were constructed by *parABCDE* amplification using pOGG99_par as a template and primers par_fw and par_rv with high-fidelity Phusion polymerase (NEB) and insertion into pCDF1b, pHERD99's, and pSEVA251-99's KpnI restriction site, respectively. This *parABCDE* operon is 99.96% identical to RP4's with a single nucleotide mismatch in the non-coding area between *parA* and *parB*.

**Table 2. Strains, plasmids, and primers.**

***Strains***

| Name | Shorthand | Notes | Reference |
|---|---|---|---|
| *Escherichia coli* DH5α | DH5α | | Common laboratory strain |
| *Escherichia coli* K12::mCherry | K12::mCherry | Chromosomal Km$^R$, mCherry, lacI insertion | [52] |
| *Escherichia coli* DH5α::CmR | DH5α::CmR | Chromosomal *cat* insertion | *This study* |
| *Escherichia coli* DH5α::GmR | DH5α::GmR | Chromosomal *aacC1* insertion | *This study* |
| *Escherichia coli* DH5α::SmR | DH5α::SmR | Chromosomal *aad* insertion | *This study* |

***Plasmids***

| Plasmid | Resistance and Payload | Notes | Reference |
|---|---|---|---|
| pKJK5::csg[aphA99] | Tetracycline, Trimethoprim, CRISPR-Cas9 cassette targeted towards *aphA* | "CRISPR-Cas on" version of pKJK5::csg | *This study* |
| pKJK5::csg[nt2] | Tetracycline, Trimethoprim, non-targeting CRISPR-Cas9 cassette | "CRISPR-Cas off" version of pKJK5::csg. Previously referred to as pKJK5::*gfp*$^{PL}$ | [50] |
| pHERD30T | Gentamicin | pBR322 vector | [53] |
| pHERD99 | Gentamicin, includes [aphA99] target site | Genbank acc. OR900359 | *This study* |
| pHERD99_par | Gentamicin, [aphA99] target site, *parABCDE* | Genbank acc. OR900360 | *This study* |
| pCDF1b | Streptomycin | | Novagen (EMD Millipore) |
| pCDF1b_par | Streptomycin, *parABCDE* | Genbank acc. OR900361 | *This study* |
| pSEVA251 | Kanamycin | IncQ vector; Genbank acc. JX560330 | [54] |
| pSEVA251−99 | Kanamycin (targetable by pKJK5::csg[aphA99]) | Genbank acc. PV231316 | *This study* |
| pSEVA251−99_par | Kanamycin, *parABCDE* | Genbank acc. PV231317 | *This study* |
| pOPS0378 | Kanamycin, mCherry, *parABCDE* | Addgene #133229 | [51] |
| pOGG99 | Kanamycin | IncP vector; Genbank acc. PV231318 | *This study* |
| pOGG99_par | Kanamycin, *parABCDE* | Genbank acc. PV231319 | *This study* |
| RP4 | Tetracycline, Kanamycin, Ampicillin, *parABCDE* | Genbank acc. L27758 | [55] |
| pBAM1-Gm | Ampicillin, Gentamicin | Tn5 *aacC1* transposon | [48] |
| pBAM1-Sm | Ampicillin, Streptomycin | Tn5 *aad* tranposon | [48] |
| pBAM1-Cm | Ampicillin, Chloramphenicol | Tn5 *cat* transposon | [49] |

***Primers and sequences***

| Name | Sequence (5′ → 3′) |
|---|---|
| [aphA99] | AAATGGGCGCGTGATAAGGT |
| [nt2] | GTTTTCTGCCTGTCGATCCAGTTTTAGAGCTCTAAAACTGGATCGACAGGCAGAAAACATGTCGATCCA |
| par_fw | TCGGTACCTGCATGAGCTTGTGGAAGTG |
| par_rv | CAGGTACCTGCTCAACAGGTTCGCA |
| aphA99PAM_top | AGCTTAAATGGGCGCGTGATAAGGTTGGTAC |
| aphA99PAM_btm | CAACCTTATCACGCGCCCATTTA |
| pOGG0_mut_fw | TGGGCGCGTGATAAGGTGGGTCAGAGCGGC |
| pOGG0_mut_rv | TTTATAGCCATACAGATCCGCATCCATGTTGCTGTTCAGACGC |

*(Continued)*

**Table 2.** (Continued)

| *Strains* | | | |
|---|---|---|---|
| Name | Shorthand | Notes | Reference |
| pOGG0_sequence | TATTGGTGAGAATCCAGGCA | | |
| pOGG99_parRemoval_top | TGAGCCAAGCTTGCATGCCTGCAGGTCGACTCTAGAGGATCCCCGGGTACCGAGCTCGAATTCACTA | | |
| pOGG99_parRemoval_btm | GATCTAGTGAATTCGAGCTCGGTACCCGGGGATCCTCTAGAGTCGACCTGCAGGCATGCAAGCTTGGC | | |

All cloned plasmids were subjected to long-read sequencing by Plasmidsaurus using Oxford Nanopore Technology with custom analysis and annotation. Annotated sequences are deposited on Genbank with IDs listed in Table 2.

## Culture conditions

Unless otherwise stated, all strains were cultured shaking 180 rpm in LB broth or statically on LB plates at 37˚C. Where appropriate, cultures or plates were supplemented with the following antibiotics at the following concentrations: Ap—100 µg/mL ampicillin; Cm—25 µg/mL chloramphenicol; Gm—50 µg/mL gentamicin; Km—50 µg/mL kanamycin; Sm—50 µg/mL streptomycin; Tc—12 µg/mL tetracycline; Tmp—10 µg/mL trimethoprim.

## Conjugation experiments

In our two-strain competition between pKJK5 and RP4 (Fig 2), we used *E. coli* DH5α::GmR containing pKJK5::csg and *E. coli* DH5α::CmR containing RP4 as mating partners. In three-strain competitions (S2 Fig), bacterial hosts used were *E. coli* DH5α::SmR containing pKJK5::csg, *E. coli* DH5α::CmR containing RP4, and *E. coli* DH5α::GmR. In both experiments, all strains also contained empty vector pCDF1b, or pCDF1b_par for TA-off treatments. A control experiment to assess the differential effects of using Ampicillin or Kanamycin to select for RP4 (S4 Fig) was set up in the same way, with the omission of bystander vectors pCDF1b or pCDF1b_par in all strains.

In additional two-strain competitions using vectors (S3 Fig), *E. coli* DH5α::SmR containing pKJK5::csg was used as donor, and the recipient was *E. coli* DH5α::CmR containing either RP4, pHERD99, pSEVA251-99, pOGG99, or their *par*-encoding derivatives.

To turn CRISPR-Cas activity on or off, we used two different pKJK5::csg variants with fixed targeting specificity: pKJK5::csg[aphA99] was programmed to target RP4's *aphA* gene, and pKJK5::csg[nt2] encoded a non-functional sgRNA with a random 69-nt target sequence that is not present in the model system (plasmid pKJK5::*gfp*^PL; [50]).

Single colonies from streak plates of mating partners were suspended in 15–25 mL LB with antibiotics appropriate to maintain their plasmid content (see Table 2) and grown overnight. These T0 cultures were then washed twice with 0.9% (w/v) NaCl and adjusted to OD600 = 0.5. Washed, OD-adjusted cultures were filter-mated in a 1:1(:1) ratio:

Filter matings were carried out using a 12-stream Millipore vacuum pump, sterilized with 70% Ethanol and UV light before and after each batch of filter mating and assembled in a Cat2 biosafety cabinet. For each mating, a 0.22 µm glass microfibre filter (Whatman) was placed onto a vacuum pump position, dampened with 200 µL sterile 0.9% NaCl, and topped with a 0.22 µm cyclopore membrane (Whatman). Fully assembled filter positions were equilibrated by applying a vacuum until 2 mL of 0.9% NaCl were pumped through. Next, 1 mL of OD-adjusted donors and 1 mL of OD-adjusted recipients (or 600 µL of each mating partner in 3-strain competitions) were added to each filter position together with 1 mL 0.9% NaCl, and a vacuum applied until the strain mix was pumped through (*N* = 5). Control matings always included donor-only, recipient-only (and naïve-strain-only where appropriate), and buffer-only sterility controls and yielded results as expected. Cyclopore membranes were placed cell-side-up onto a 10% LB plate (supplemented with 0.9% NaCl) and incubated at 37 °C for 36 hours. Then, cells were recovered by placing each filter into 3 mL of 0.9% NaCl and vortexed for 15 s. This cell

suspension was frozen in 20% (w/v) glycerol at −70 °C and plated onto selective plates with various antibiotic regimes to assess strain prevalence and strains' plasmid content.

## Selective plating

Samples were plated in triplicate in 5 μL droplets in dilutions ranging from $10^0$ to $10^{-7}$ onto LB medium containing different combinations of antibiotics. Cm and Gm were used to select for DH5α::CmR and DH5α::GmR, respectively, and combining these antibiotics with Tmp and Km allowed growth of only DH5α hosts carrying pKJK5 and RP4, respectively. Colonies were counted at the most appropriate dilution, and GFP fluorescence of pKJK5::csg-containing colonies confirmed stringency of selective regimes.

To assess overall sample composition and plasmid content of the DH5α::SmR host, we plated 50 μL of a $10^{-5}$ dilution of each replicate onto LB plates without selection and suspended 96 colonies of each sample in 250 μL LB each following overnight incubation. Previous assessment of colonies from selective plates sub-cultured on non-selective media indicated that RP4 and pKJK5::csg did not become lost over this timeframe in the absence of selection. The 96 clones picked from each replicate were then replica-stamped onto plates containing no selection, Cm, Gm, Km, and/or, Tmp. Clones growing on LB, but not LB supplemented with Cm or with Gm were scored as DH5α::SmR clones, and the Km/Tmp selective plates were used to score their plasmid content.

Resistance to Km is conferred by *aphA*, which is the target of the CRISPR spacer on pKJK5::csg. Plasmid of escape of CRISPR-Cas by target mutation is unusual [56], and in a control experiment, we accordingly found selection using Km to be robust to assess RP4 presence (S4 Fig).

## Plasmid competitive ratio

We calculated pKJK5 and RP4's competitive ratio (log odds ratio) as follows:

$$\text{competitive ratio} = \log \left( \frac{\text{hosts containing pKJK5}}{\text{hosts containing RP4}} \right)$$

This gives a measure of which plasmid is more frequent in a certain host and therefore won the competition for the host at the end of the experiment. Positive competitive ratios indicate pKJK5 winning the competition, negative ratios indicate RP4 winning, and ratios ≈ 0 indicate equal carriage of pKJK5 and RP4 in the host.

## Statistical analyses

Colony counts on all selective plates from all experiments, together with plasmid competitive ratios used to produce figures are available in S1 Data.

Data processing, data visualization, and statistical analyses were carried out using R version 4.3.2 and RStudio version 2023.09.1 + 494 with the following packages: flextable, dplyr, readr, tidyr, ggplot2.

**Competitive ratio.** We carried out one-sample T tests and Bonferroni adjustment for multiple testing to test for significant differences of plasmid competitive ratio to 0, indicating either RP4 or pKJK5 winning the plasmid competition. Statistical significance of 'TA off' competitive ratio in DH5α::SmR could not be assessed due to $N = 1$, as no RP4 carriage was recorded in most replicates. For all *p* values, see Tables 3 and 4.

In addition, we analyzed the contribution of CRISPR-Cas and TA status to the plasmid competition outcome by fitting separate Generalized Linear Models to data generated in the two-strain competition. We chose Gaussian GLMs with identity link function, and modeled competitive ratio as a function of CRISPR-Cas status, TA status, and their interaction. The inclusion of additional variables (e.g., replicate and position on filter pump) was tested, and the final models chosen for not including non-significant explanatory variables and upholding model assumptions. We investigated the modeling coefficients (Table 5) to determine whether these variables have a significant influence on plasmid competition outcome.

**Table 3. T tests adjusted for multiple testing (Fig 2). Significant values after Bonferroni adjustment in bold. CRISPR_action refers to *Escherichia coli* DH5α chromosomal tag identity; defence—GmR (defensive CRISPR-Cas), offence—CmR (offensive CRISPR-Cas).**

| CRISPR | TA | CRISPR_action | p value (unadjusted) |
|--------|-----|---------------|----------------------|
| off | off | defence | **0.00114** |
| off | off | offence | 0.0345 |
| off | on | defence | 0.0872 |
| off | on | offence | 0.0458 |
| on | off | defence | **0.000000992** |
| on | off | offence | **0.000959** |
| on | on | defence | **0.00000177** |
| on | on | offence | **0.0000140** |

Bonferroni-adjusted significance threshold $\alpha = 0.0063$

**Table 4. T tests adjusted for multiple testing (S2 Fig). Significant values after Bonferroni adjustment in bold. Host refers to *Escherichia coli* DH5α chromosomal tag identity; S—SmR (defensive CRISPR-Cas), C—CmR (offensive CRISPR-Cas), G—GmR (naïve host).**

| CRISPR-Cas status | host | TA status | p value (unadjusted) |
|-------------------|------|-----------|----------------------|
| on | C | on | **0.0000366** |
| on | C | off | 0.0720 |
| on | G | on | 0.0967 |
| on | G | off | **0.00186** |
| on | S | on | **0.000000261** |
| off | C | on | **0.000998** |
| off | C | off | 0.469 |
| off | G | on | **0.000137** |
| off | G | off | 0.266 |
| off | S | on | 0.185 |

Bonferroni-adjusted significance threshold $\alpha = 0.005$

Two-strain competitions with vectors showed different baseline competitive ratios dependent on vector identity. Therefore, we analyzed relative differences in competitive ratio between treatments by fitting separate Generalized Linear Models to data generated in two-strain competitions (S3 Fig). For model structures and coefficients see Table 6. RP4 (with bystander), pOGG99, pHERD99, and pSEVA251−99 data were followed up by Tukey's HSD test, see Table 7.

**Streptomycin-resistant c.f.u.** The log10 of the proportion of streptomycin-resistant c.f.u. presented in S1 Fig followed a normal distribution and was assessed by one-sample *T* tests and Bonferroni adjustment for multiple testing for differences to 0 (log10 of 1; complete maintenance of bystander vector pCDF1b). No differences were statistically significant (Table 8).

## Bioinformatic analysis of plasmid TA carriage

The sequences of 17,828 de-replicated plasmid found across prokaryotic genomes [6] were retrieved from NCBI using rentrez [57]. CRISPR-Cas presence and targeting information were adopted from [6]. Plasmid open reading frames were predicted with prodigal v2.6.3 [58]. TA nucleotide or protein sequences were retrieved from TADB v3.0 [14]. *parDE* and other well-studied PSK systems were identified manually based on a recent review [13]. To identify complete TA systems, plasmids were searched with BLAST v2.16.0 [59] and hits were filtered for cases where both a toxin and antitoxin

**Table 5. GLM structures and coefficients for Fig 2.**

| Defensive CRISPR-Cas | comprat~CRISPR_status*TA_status | | | | |
|---|---|---|---|---|---|
| | Estimate | Standard Error | *t* value | Pr(>\|z\|) | |
| (Intercept) | 0.25456 | 0.08595 | 2.962 | 0.00918 | |
| CRISPR_statusON | 4.79264 | 0.12156 | 39.427 | <2e-16 | ** |
| TA_statusON | −0.10573 | 0.12156 | −0.870 | 0.39726 | *** |
| CRISPR_statusON:TA_statusON | 0.11083 | 0.17191 | 0.645 | 0.52826 | |

*Signif. codes: 0 <= '***' < 0.001 < '**' < 0.01 < '*' < 0.05*

(Dispersion parameter for gaussian family taken to be 0.03439149)

Null deviance: 118.13714 on 19 degrees of freedom

Residual deviance: 0.61905 on 18 degrees of freedom

| Offensive CRISPR-Cas | comprat~CRISPR_status*TA_status | | | | |
|---|---|---|---|---|---|
| | Estimate | Standard Error | *t* value | Pr(>\|z\|) | |
| (Intercept) | 0.4119 | 0.1204 | 3.422 | 0.003497 | ** |
| CRISPR_statuson | 0.7267 | 0.1703 | 4.268 | 0.000588 | *** |
| TA_statusON | −0.7354 | 0.1703 | −4.319 | 0.000529 | *** |
| CRISPR_statusON:TA_statusON | −3.0803 | 0.2408 | −12.793 | 8.1e-10 | *** |

*Signif. codes: 0 <= '***' < 0.001 < '**' < 0.01 < '*' < 0.05*

(Dispersion parameter for gaussian family taken to be 0.0724714)

Null deviance: 42.2187 on 19 degrees of freedom

Residual deviance: 1.1595 on 16 degrees of freedom

component were directly adjacent. For the analysis of TA systems carried by CRISPR-Cas bearing plasmids, full TA systems were considered alongside orphan antitoxin genes, and a >70% alignment coverage in comparison with TADB was considered as a match. Counts of CRISPR loci, Cas loci, TA systems, and the network of TA families on plasmids carrying or targeted by CRISPR-Cas systems are available in S1 Data. Statistical analyses were run using R v4.4.2 base functions.

## Supporting information

**S1 Fig. Bystander plasmids were maintained.** Mean ± standard error of the proportion of Streptomycin resistant Colony Forming Units (CFU) after mating was never significantly different from 1 (*T* test followed by Bonferroni adjustment for multiple testing; Table 7). Additionally, when stamp plating, only two individual clones out of 2,574 were found to be streptomycin sensitive, one of which was from a single-host control treatment. Vector presence was confirmed by PCR in all tested DH5α::SmR colonies. The data underlying this Figure can be found in S1 Data; the competitive outcome for this experiment is presented in S2 Fig.
(TIF)

**S2 Fig. Plasmid competition results were upheld in a three-strain model system. (A)** CRISPR-Cas and TA activity were tweaked on competitor plasmids in a binary manner by switching each system on or off. pKJK5::csg carries a gene cassette encompassing *cas9, sgRNA, GFP* with either an RP4-targeting or non-targeting guide, RP4 naturally carries TA system *parABCDE*. **(B)** Plasmids were competed using three *Escherichia coli* DH5α hosts with different plasmid content, allowing us to assess competitive outcome under defensive, offensive, or no clear mode of CRISPR-Cas action. After filter mating, plasmid content of each host was assessed by selective plating. **(C–E)** Mean ± standard error of the competitive ratio (log odds ratio of pKJK5-carrying hosts/RP4-carrying hosts) describes outcome of plasmid competition (*N* = 5). Values >0 indicate CRISPR-Cas plasmid pKJK5 winning the competition, and values <0 indicate TA plasmid RP4 winning the

**Table 6. GLM structures and coefficients for S3 Fig.**

| RP4 with bystander model | comprat~CRISPR_status*TA_status | | | | |
|---|---|---|---|---|---|
| | **Estimate** | **Standard Error** | **z value** | **Pr(>\|z\|)** | |
| (Intercept) | − 0.863 | 0.198 | − 4.359 | 0.0024 | ** |
| CRISPR_statusON | 1.554 | 0.280 | 5.550 | 0.0005 | *** |
| TA_statusON | −0.207 | 0.280 | −0.738 | 0.4815 | – |
| CRISPR_statusON:TA_statusON | −6.603 | 0.396 | −16,676 | 0.0000 | *** |

*Signif. codes: 0 <= '***' < 0.001 < '**' < 0.01 < '*' < 0.05*

(Dispersion parameter for gaussian family taken to be 0.1175985)

Null deviance: 79.73 on 11 degrees of freedom

Residual deviance: 0.9408 on 8 degrees of freedom

| RP4 model | comprat~CRISPR_status | | | | |
|---|---|---|---|---|---|
| | Estimate | Standard Error | z value | Pr(>\|z\|) | |
| (Intercept) | −0.150 | 0.062 | −2.409 | 0.0737 | . |
| CRISPR_statuson | −2.695 | 0.088 | −30.687 | 0.0000 | *** |

*Signif. codes: 0 <= '***' < 0.001 < '**' < 0.01 < '*' < 0.05*

(Dispersion parameter for gaussian family taken to be 0.0115664)

Null deviance: 10.94 on 5 degrees of freedom

Residual deviance: 0.04627 on 4 degrees of freedom

| pOGG99 model | comprat~CRISPR_status*TA_status | | | | |
|---|---|---|---|---|---|
| | Estimate | Standard Error | z value | Pr(>\|z\|) | |
| (Intercept) | −2.180 | 0.190 | −11.486 | 0.0000 | *** |
| CRISPR_statusON | 2.762 | 0.268 | 10.289 | 0.0000 | *** |
| TA_statusON | 0.008 | 0.268 | 0.030 | 0.9765 | |
| CRISPR_statusON:TA_statusON | −3.633 | 0.380 | −9.570 | 0.0000 | *** |

*Signif. codes: 0 <= '***' < 0.001 < '**' < 0.01 < '*' < 0.05*

(Dispersion parameter for gaussian family taken to be 0.108098)

Null deviance: 23.26 on 11 degrees of freedom

Residual deviance: 0.8648 on 8 degrees of freedom

| pHERD99 model | comprat~CRISPR_status*TA_status | | | | |
|---|---|---|---|---|---|
| | Estimate | Standard Error | z value | Pr(>\|z\|) | |
| (Intercept) | 0.488 | 0.134 | 3.649 | 0.0065 | ** |
| CRISPR_statusON | 3.360 | 0.189 | 17.769 | 0.0000 | *** |
| TA_statusON | −0.753 | 0.189 | −3.982 | 0.0041 | ** |
| CRISPR_statusON:TA_statusON | −3.200 | 0.267 | −11.970 | 0.0000 | *** |

*Signif. codes: 0 <= '***' < 0.001 < '**' < 0.01 < '*' < 0.05*

(Dispersion parameter for gaussian family taken to be 0.05362118)

Null deviance: 34.01 on 11 degrees of freedom

Residual deviance: 0.429 on 8 degrees of freedom

| pSEVA251-99 model | comprat~CRISPR_status*TA_status | | | | |
|---|---|---|---|---|---|
| | Estimate | Standard Error | z value | Pr(>\|z\|) | |
| (Intercept) | 0.056 | 0.127 | 0.440 | 0.6717 | |
| CRISPR_statusON | 0.218 | 0.180 | 1.211 | 0.2603 | |
| TA_statusON | −0.234 | 0.180 | −1.297 | 0.2308 | |
| CRISPR_statusON:TA_statusON | −2.475 | 0.255 | −9.710 | 0.0000 | *** |

*Signif. codes: 0 <= '***' < 0.001 < '**' < 0.01 < '*' < 0.05*

*(Continued)*

**Table 6.** (Continued)

| RP4 with bystander model | comprat~CRISPR_status*TA_status | | | | |
|---|---|---|---|---|---|
| | **Estimate** | **Standard Error** | **z value** | **Pr(>|z|)** | |
| (Dispersion parameter for gaussian family taken to be 0.04870928) | | | | | |
| Null deviance: 14.59 on 11 degrees of freedom | | | | | |
| Residual deviance: 0.3897 on 8 degrees of freedom | | | | | |

**Table 7. Tukey's HSD results for S3A, S3C–S3E Fig.**

| RP4 with bystander (CRISPR:TA status) | diff | lwr | upr | p value (adjusted) |
|---|---|---|---|---|
| off:off-on:on | 5.2557215 | 4.3590686 | 6.152374 | **2.9E−07** |
| off:on-on:on | 5.0490158 | 4.1523629 | 5.945669 | **4.2E−07** |
| on:off-on:on | 6.8098491 | 5.9131962 | 7.706502 | **3.2E−08** |
| off:on-off:off | 0.2067057 | − 0.6899471 | 1.103359 | 8.8E−01 |
| on:off-off:off | 1.5541276 | 0.6574748 | 2.450780 | **2.4E−03** |
| on:off-off:on | 1.7608333 | 0.8641805 | 2.657486 | **1.1E−03** |
| **pOGG99** (CRISPR:TA status) | diff | lwr | upr | p value (adjusted) |
| off:off-on:on | 0.863006847 | 0.003335675 | 1.7226780 | **4.9E−02** |
| off:on-on:on | 0.871163151 | 0.011491979 | 1.7308343 | **4.7E−02** |
| on:off-on:on | 3.625170716 | 2.765499544 | 4.4848419 | **4.1E−06** |
| off:on-off:off | 0.008156304 | −0.851514868 | 0.8678275 | 9.9E−01 |
| on:off-off:off | 2.762163869 | 1.902492697 | 3.6218350 | **3.2E−05** |
| on:off-off:on | 2.754007565 | 1.894336393 | 3.6136787 | **3.3E−05** |
| **pHERD99** (CRISPR:TA status) | diff | lwr | upr | p value (adjusted) |
| on:on-off:on | 0.1590421 | −0.44642645 | 0.7645107 | 8.3E−01 |
| off:off-off:on | 0.7528541 | 0.14738553 | 1.3583227 | **1.7E−02** |
| on:off-off:on | 4.1123653 | 3.50689671 | 4.7178339 | **7.5E−08** |
| off:off-on:on | 0.5938120 | −0.01165662 | 1.1992806 | 5.5E−02 |
| on:off-on:on | 3.9533232 | 3.34785456 | 4.5587918 | **1.1E−07** |
| on:off-off:off | 3.3595112 | 2.75404258 | 3.9649798 | **4.7E−07** |
| **pSEVA251−99** (CRISPR:TA status) | diff | lwr | upr | p value (adjusted) |
| off:on-on:on | 2.2563024 | 1.6792313 | 2.8333734 | **7.3E−06** |
| off:off-on:on | 2.4900116 | 1.9129406 | 3.0670827 | **3.5E−06** |
| on:off-on:on | 2.7082955 | 2.1312244 | 3.2853665 | **1.8E−06** |
| off:off-off:on | 0.2337093 | −0.3433617 | 0.8107803 | 5.9E−01 |
| on:off-off:on | 0.4519931 | −0.1250779 | 1.0290642 | 1.3E−01 |
| on:off-off:off | 0.2182838 | −0.3587872 | 0.7953549 | 6.4E−01 |

competition for a certain host. The dashed line indicates a neutral outcome with competitive ratio = 0. Data are presented for treatments in which CRISPR-Cas and TA activity were toggled on or off in all combinations. Opacity of datapoints indicate number of replicates used to calculate means, this affects panel C TA-off data ($N=1$) and panel C TA-on and CRISPR-Cas-on data ($N=4$). Stars indicate significant differences from 0 as assessed by $T$ test and Bonferroni adjustment for multiple testing; *$p < 0.005$ with $\alpha = 0.005$; see Table 4 for all $p$ values. na -not assessed due to $N=1$ after removal

**Table 8.** *T* tests adjusted for multiple testing (S1 Fig). No values were significant after Bonferroni adjustment.

| CRISPR status | TA status | *p* value (unadjusted) |
|---|---|---|
| off | off | 0.543 |
| off | on | 0.801 |
| on | off | 0.0414 |
| on | on | 0.985 |

Bonferroni-adjusted significance threshold $\alpha = 0.0125$

of missing values (no carriage of TA plasmid RP4 recorded in 4 out of 5 replicates). The data underlying this Figure can be found in S1 Data. See S1 Text for additional information.
(TIF)

**S3 Fig. Outcome of competition with cloning vectors.** Mean ± standard error of the competitive ratio (log10 of pKJK5-carrying hosts/competitor-carrying hosts) describes the outcome of plasmid competition (*N* = 5). Values >0 indicate CRISPR-Cas plasmid pKJK5 winning the competition, and values <0 indicate the TA competitor plasmid winning the competition for a DH5α host where CRISPR-Cas was acting offensively. Data are presented for treatments in which CRISPR-Cas and TA activity were toggled on or off in all combinations. RP4 competition outcome **(A, B)** in this model system matched the outcome observed earlier (Fig 2B). Relative differences between treatments are indicated with gray lines and were assessed by Tukey's HSD after fitting individual Generalized Linear Models; see Methods and Tables 6 and 7 for model details and *p* values. *$p < 0.05$; ***$p < 0.001$; n.s. not significant $p > 0.63$. The data underlying this Figure can be found in S1 Data. See S1 Text for additional information.
(TIF)

**S4 Fig. Assessed RP4 proportion is largely robust to antibiotic used to select for RP4.** Mean and standard error of RP4 content of hosts in pilot mating experiment lacking a bystander plasmid assessed by ampicillin or by kanamycin, *N* = 5. Data are presented for CRISPR-Cas switched on or off. The data underlying this Figure can be found in S1 Data. See S1 Text for additional information.
(TIF)

**S5 Fig. Plasmid-borne CRISPR-Cas systems and their targets. (A)** CRISPR-Cas systems encoded on plasmids; split into distribution of Cas operon and CRISPR array types. Type IV systems are highlighted to emphasize their predominance. **(B)** The total number of plasmids encoding each TA family, highlighting *parDE* and other well-studied post-segregational killing (PSK) systems [13]. The other bar charts show the proportion of plasmids encoding each TA family targeted by plasmid borne Type IV and non-Type IV CRISPR-Cas systems. The data underlying this Figure can be found in S1 Data.
(TIF)

**S6 Fig. Contingency tables of expected versus observed values for the association between plasmid-borne CRISPR-Cas targeting and TA carriage.** Contingency tables showing the relationship between plasmid targeting and TA presence. The tables display the observed counts of plasmids with or without TA systems and whether they are targeted or not by CRISPR spacers in another plasmid, either for all CRISPR-Cas types **(A)**, Type IV CRISPR-Cas only **(B)**, or for non-Type IV CRISPR-Cas **(C)**. The bar plots show the sum of counts for each vertical or horizontal group. Facets in the table are coloured by standardized residual after Pearson's $\chi^2$ test, indicating magnitude and directionality of association (red—less than expected, blue—more than expected). The data underlying this Figure can be found in S1 Data.
(TIF)

**S1 Text. Supplementary results.** Additional narrative and data analysis of S2–S4 Figs. Three-strain competition showed putative generalisability of CRISPR-Cas detriment in offense when facing TA (S2 Fig), the competitive outcome was not due to RP4's backbone (S3 Fig), and measured RP4 proportions were robust to altering the selective agent to Ampicillin (S4 Fig).
(DOCX)

**S2 Text. Supplementary mathematical analysis.** Development of the mathematical model (Fig 1) including derivation of equations used, and mathematical analysis of the impact of changing competitive traits $x$ and $y$ on plasmid transition rates and the outcome of plasmid competition.
(DOCX)

**S1 Data. Supplementary data to CRISPR-Cas is beneficial in plasmid competition, but limited by competitor toxin–antitoxin activity when horizontally transferred.**
(XLSX)

## Acknowledgments

The authors would like to thank Mario Mestre (KU Copenhagen, Denmark) and Edze Westra (University of Exeter, UK) for stimulating and inspiring discussions and enthusiasm. The authors further thank Iolanda Domingues (Exeter Centre for Cytomics, University of Exeter, UK) for project scoping using flow cytometry.

For the purpose of open access, the author has applied a 'Creative Commons Attribution (CC BY)' licence to any Author Accepted Manuscript version arising from this submission.

## Author contributions

**Conceptualization:** David Sünderhauf, Sam P. Brown, Stineke van Houte.

**Data curation:** David Sünderhauf, Jahn R. Ringger.

**Formal analysis:** David Sünderhauf, Leighton J. Payne, Sam P. Brown.

**Funding acquisition:** William H. Gaze, Stineke van Houte.

**Investigation:** David Sünderhauf, Jahn R. Ringger, Leighton J. Payne, Rafael Pinilla-Redondo, Sam P. Brown.

**Methodology:** David Sünderhauf, Leighton J. Payne, Rafael Pinilla-Redondo.

**Project administration:** David Sünderhauf.

**Supervision:** David Sünderhauf, William H. Gaze, Stineke van Houte.

**Validation:** David Sünderhauf, Rafael Pinilla-Redondo.

**Visualization:** David Sünderhauf.

**Writing – original draft:** David Sünderhauf.

**Writing – review & editing:** David Sünderhauf, Jahn R. Ringger, Leighton J. Payne, Rafael Pinilla-Redondo, William H. Gaze, Sam P. Brown, Stineke van Houte.

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
