## [Editor Report · Decision Letter 0]

6 Jun 2025

Dear Dr Sünderhauf,

Thank you for submitting your manuscript entitled "CRISPR-Cas is beneficial in plasmid competition, but limited by competitor toxin-antitoxin activity when horizontally transferred." for consideration as a Research Article by PLOS Biology.

Your manuscript has now been evaluated by the PLOS Biology editorial staff, as well as by an academic editor with relevant expertise, and I am writing to let you know that we would like to send your submission out for external peer review.

Once your full submission is complete, your paper will undergo a series of checks in preparation for peer review. After your manuscript has passed the checks it will be sent out for review. To provide the metadata for your submission, please Login to Editorial Manager (https://www.editorialmanager.com/pbiology) within two working days, i.e. by Jun 08 2025 11:59PM.

Kind regards,

Melissa

Melissa Vazquez Hernandez, Ph.D.

Associate Editor

PLOS Biology

---

## [Decision Letter · Decision Letter 1]

21 Jul 2025

Dear Dr Sünderhauf,

Thank you for your patience while your manuscript "CRISPR-Cas is beneficial in plasmid competition, but limited by competitor toxin-antitoxin activity when horizontally transferred." was peer-reviewed at PLOS Biology. It has now been evaluated by the PLOS Biology editors, an Academic Editor with relevant expertise, and by two independent reviewers.

In light of the reviews, which you will find at the end of this email, we would like to invite you to revise the work to thoroughly address the reviewers' reports. As you will see below, while one reviewer is positive, another reviewer will require clarifications. Reviewer 1 thinks the question is important but that some of the central claims might be undermined by misinterpretations by the model and the biological data. Specifically, the reviewer mentions the discrepancies between the model and the biological observations like the effect of CRISPR-Cas when TA is present, and then the derived bioinformatic analyses that are based on the previous conclusions. Reviewer 2 questions some of the parameters used in the model, as well as how other competitive strategies may affect the conclusions regarding CRISPR vs TA systems. As Reviewer 1. s/he also mentions that Type IV CRISPR should be considered in the bioinformatic analysis.

IMPORTANT: following discussions with the reviewers and the Academic Editor, we believe that all concerns can be addressed through adjustments to the model, refinements to the comparative analyses, and clarifications within the manuscript. It may be possible that some concerns stem from misunderstandings, and we believe that implementing Reviewer 2’s suggestions will help resolve some of Reviewer 1’s points as well.

Given the extent of revision needed, we cannot make a decision about publication until we have seen the revised manuscript and your response to the reviewers' comments. Your revised manuscript is likely to be sent for further evaluation by all or a subset of the reviewers.

**IMPORTANT - SUBMITTING YOUR REVISION**

*Re-submission Checklist*

*Published Peer Review*

*PLOS Data Policy*

*Blot and Gel Data Policy*

Sincerely,

Melissa

Melissa Vazquez Hernandez, Ph.D.

Associate Editor

PLOS Biology

REVIEWERS' COMMENTS

Reviewer #1:

The authors investigate the competition between a plasmid carrying a CRISPR-Cas defense system and one carrying a toxin-antitoxin (TA) system. This is a biologically interesting scenario, as CRISPR-Cas systems can target and degrade competing plasmids, while TA systems typically induce cell death upon plasmid loss, due to the greater stability of toxins relative to antitoxins. To explore this, the authors construct a toy model of plasmid competition, validate predictions with experiments, and perform a bioinformatic analysis of plasmid co-occurrence.

The manuscript is generally well-written, the figures are clear, and the experimental data are presented effectively. Although the dataset is limited in size, I do not view this as problematic provided the conclusions are robust and well-supported. However, I find several key interpretations unconvincing, both in terms of the modeling and the biological conclusions. If my concerns are valid, they undermine the central claims of the manuscript. I would welcome clarification, but in its current form I do not consider the work suitable for publication.

1. Discrepancies between model predictions and experimental results:

The authors assert strong agreement between theoretical predictions and experimental results. However, several inconsistencies suggest otherwise:

In Fig. 2C, the CRISPR-Cas system has a strong effect when TA is present and a weak or no effect when TA is absent. However, the model (Fig. 1B) predicts the opposite: the effect of CRISPR-Cas (measured as the difference between CRISPR-Cas strength = 0 vs >0) is greatest when TA is absent. This contradicts the experimental results.

The authors claim that CRISPR-Cas is highly detrimental in the presence of a resident TA plasmid, and this is observed experimentally (Fig. 2D). However, the model (Fig. 1C, black line) shows only a small decrease in survival, suggesting minimal detrimental effect. On closer inspection, the model implies that CRISPR-Cas is always beneficial for the resident plasmid, especially when the invading plasmid lacks or has a weak TA system. Overall, the dominant factor appears to be whether TA is present or absent, rather than the presence of CRISPR-Cas.

2. Interpretation and implementation of the bioinformatic analysis:

The authors hypothesize that co-existence between TA and CRISPR-Cas systems should be rare due to the model's predicted competition, and they investigate this via a plasmid database.

As discussed above, I do not believe the model robustly predicts a strong anti-correlation, undermining the rationale for this hypothesis.

The exclusion of Type IV CRISPR systems from the analysis is insufficiently justified. Although Type IV systems are thought to suppress gene expression rather than cleave DNA, suppression of antitoxin expression would likely still trigger toxin-mediated cell death. Reference [8], which the authors cite to justify this exclusion, does not appear to address TA systems and does not support this decision.

Most critically, the model and experiments do not permit co-existence of TA and CRISPR-Cas plasmids — they are framed in terms of competition between mutually exclusive plasmids. Yet the bioinformatic analysis examines co-occurrence in the same cell. As such, the model and experiments do not directly inform expectations for plasmid co-existence in natural populations. Furthermore, CRISPR-Cas systems would not be expected to target co-resident TA plasmids, which would presumably be recognized as self. Thus, the connection between the first two figures and the database analysis is tenuous.

Conclusion:

In summary, while the study addresses a compelling biological question and employs an integrative approach, I find major issues in the interpretation of both the model and the bioinformatic analysis. These concerns raise doubts about the central conclusions of the work. Unless the authors can convincingly resolve these points, I do not consider the manuscript suitable for publication.

Reviewer #2:

The authors explore the benefits of carrying CRISPR-Cas and TA systems in plasmid competition using mathematical modelling, experiments and bioinformatic analysis. The study shows that the benefit of competition systems is dependent on the ecological context, invader versus resident. CRISPR-Cas was shown to be a generally beneficial strategy for resident plasmids but become detrimental when trying to invade a population of TA-system carrying plasmids. The bioinformatic study suggests that these effects are dependent on the CRISPR-Cas type.

The study of plasmid competition and interactions is timely and exciting. Overall, the combination of experiments, modelling and bioinformatic analysis comprehensively supports the conclusions. The manuscript is well written, and I would like to particularly applaud the diligent methods description.

Major comments:

IncP1 plasmids contain entry exclusion systems and belong to the same plasmid incompatibility group. This will drastically lower the rate of coinfection as compared to non-related plasmids and potentially lead to heterogeneity in plasmid carriage within the population. How does the addition of these competitive strategies affect the conclusions about CRISPR versus TA systems? It seems that this is being addressed to some extent in Fig. S4, but this needs more explanation and the comparison with Fig. 2 is made a bit difficult because E. coli K12 was used as a donor instead of E. coli DH5a. Conjugation rates with these two donors could vary substantially, given that DH5a has a reduced genome making it more suitable for molecular cloning and transformation.

While I agree with the general idea behind the parameter f and the potential benefit of PSK for surrounding cells if cell lysis occurs, I am not entirely sure that it is justified for this study. The experimental system is using a TA system that works through DNA gyrase inhibition, which results in a more bacteriostatic effect and, if lysis occurs at all, probably with a substantial time delay. Rather, these TA systems can lead to SOS response activation and cell filamentation, which would result in a neutral to negative effect on surrounding cells.

There are a few interesting points in the bioinformatic analysis that might be worth exploring further and would strengthen the argument:

First, it wasn't mentioned if CRISPR-Cas carrying plasmids do also carry TA systems? Or potentially they even just carry anti-toxins?

Second, it looks like Type IV CRISPR targeting of TA carrying plasmid is higher than expected. It would be interesting to see if Type IV CRISPR target specifically the toxin gene of the TA system, so that segregational loss of the plasmid is not detrimental anymore.

Third, the authors investigated CRISPR-Cas types, but looking also into TA system types and their potential to lyse cells (rapidly) due to PSK could indicate the relevance of actual lysis or just cell death.

Selective plating for RP4 using kanamycin, which is also the target of the CRISPR spacer, might be misleading if the plasmids lose kanamycin to escape CRISPR but still manage to persist.

Minor comments:

L27: Given the title and the following sentence it is a bit unclear here if that is a general statement and how it relates to TA systems. The TA systems then come out of nowhere in the next sentence.

L29: maybe 'defends its host' might be clearer than 'remains' ('remains' might trigger the question, as opposed to doing what?)

L53: this makes it sound like the difference between chromosomal versus MGE encoded immune systems will be investigated

L88: host population

L367-381: it is a bit unclear here how co-opting the host's spacer machinery will help in overcoming TA activity

Figure 1 and SI: What is the benefit of adding the baseline fitness parameter a and transition rate scaling parameter b to the fitness analysis as they do not seem to be used in this study and are always set to 1?

Fig. S4: it looks like the lower part of panel B is cut off?

---

## [Decision Letter · Decision Letter 2]

8 Dec 2025

Dear Dr Sünderhauf,

Thank you for your patience while we considered your revised manuscript "CRISPR-Cas is beneficial in plasmid competition, but limited by competitor toxin-antitoxin activity when horizontally transferred." for publication as a Research Article at PLOS Biology. Your revised study has been evaluated by the PLOS Biology editors, the Academic Editor and the original reviewers.

In light of the reviews, which you will find below, we would like to invite you to submit a Major Revision that thoroughly addresses the remaining methodological concerns. As you will see, Reviewer #2 is now satisfied with the revisions and supports acceptance. However, Reviewer #1 and the Academic Editor continue to raise substantive issues regarding the coherence between the model predictions and the experimental data, particularly in Figure 2. Reviewer #1 highlights what seems like a fundamental methodological inconsistency in the delineation between “core” and “peripheral” predictions and notes that the central claim lacks sufficient statistical support in the current dataset. In addition, both Reviewer #1 and the Academic Editor note that the presentation of the data in Figure 2C–E is difficult to interpret, and that the unconventional visualization (e.g., the use of opacity to denote replicate numbers) may be contributing to misunderstandings and concerns about the treatment of conflicting data that does not align with the model predictions. To move forward, we will require clearer data presentation in Figure 2 as well as additional experimental support, like increasing sample number, to robustly test the major claims.

IMPORTANT: We agree with these concerns and consider that addressing them experimentally will substantially strengthen the manuscript. The revised version will be returned to Reviewer #1 for reassessment and, if needed, may also be evaluated by an additional reviewer to provide an independent perspective on the statistical and interpretive issues surrounding Figure 2, although we will add such a reviewer only if strictly necessary.

Given the extent of revision needed, we cannot make a decision about publication until we have seen the revised manuscript and your response to the reviewers' comments. Your revised manuscript is likely to be sent for further evaluation by all or a subset of the reviewers.

**IMPORTANT - SUBMITTING YOUR REVISION**

*Re-submission Checklist*

*Published Peer Review*

*PLOS Data Policy*

*Blot and Gel Data Policy*

Sincerely,

Melissa

Melissa Vazquez Hernandez, Ph.D.

Associate Editor

PLOS Biology

REVIEWERS' COMMENTS

Reviewer #1:

The authors have satisfactorily addressed several of my initial concerns, most notably the motivation behind the exclusion of Type IV systems and the subsequent design of the bioinformatic analysis, which I appreciate the clarification on.

However, a fundamental methodological inconsistency remains regarding the discrepancies between the model predictions and experimental results, specifically concerning the authors' defense that these discrepancies are confined to "peripheral predictions."

The current delineation between "core" and "peripheral" predictions lacks a clear, a priori justification, making the framework feel retrospective. This is particularly problematic when the primary claim established in the title is categorized in a way that allows contradictory experimental evidence to be discounted.

The first part of the title claims that "crispr-cas is beneficial in plasmid competition," which constitutes a central, core hypothesis of the manuscript. Yet, the experimental results presented in Figure 2C and 2D (specifically the square data points) clearly do not observe this beneficial effect, neither in offense nor in defense.

The diamond data points are consistent with the model predictions and corroborate the second half of the title. However, the lack of a significant effect observed with the square data points directly contradicts the first half of the title and what should be a core prediction.

Given this fundamental conflict between the primary hypothesis (as stated in the title) and the supporting data:if the claim from the title cannot be statistically assessed due to insufficient statistical power, the authors should increase the number of replicates to establish significance.

Reviewer #2:

The authors have improved the manuscript through changes to the model, additional experiments and further bioinformatic analyses, addressing the reviewer comments in a satisfactory manner. The simplified model analysis and improved presentation of Figure 1 substantially increase the clarity of the model comparison and the additional bioinformatic analyses provide interesting information that lend more confidence to the results.

---

## [Decision Letter · Decision Letter 3]

16 Jan 2026

Dear Dr Sünderhauf,

Thank you for your patience while we considered your revised manuscript "CRISPR-Cas is beneficial in plasmid competition, but limited by competitor toxin-antitoxin activity when horizontally transferred." for publication as a Research Article at PLOS Biology. This revised version of your manuscript has been evaluated by the PLOS Biology editors, the Academic Editor and Reviewer 1.

Based on the reviews, we are likely to accept this manuscript for publication, provided you satisfactorily address the remaining editorial points. Please also make sure to address the following data and other policy-related requests.

1) Please add to your Competing Interests the following statement “SM is a member of PLOS Biology’s Editorial Board. The other authors declare that no competing interests exist.". Please add this both in your text and the metadata during resubmission.

2) Please note that per journal policy, the model system/species studied (E. coli) should be clearly stated in the abstract of your manuscript.

3) We do not have a word limit. Please move the supplementary methods, results and references to the main text which can provide the readers an easier access to all information. Supplementary figures can of course remain as such.

4) Thank you for providing the numerical values for the figures. Could you please specify to which panel in the figures the data belongs to?

5) Please cite the location of the data clearly in all relevant main and supplementary Figure legends, e.g. “The data underlying this Figure can be found in S1 Data” or “The data underlying this Figure can be found in https://doi.org/10.5281/zenodo.XXXXX”

6) Supplementary files (e.g., excel). Please ensure that all data files are uploaded as 'Supporting Information' and are invariably referred to (in the manuscript, figure legends, and the Description field when uploading your files) using the following format verbatim: S1 Data, S2 Data, etc. Multiple panels of a single or even several figures can be included as multiple sheets in one excel file that is saved using exactly the following convention: S1_Data.xlsx (using an underscore).

7) Please ensure that your Data Statement in the submission system accurately describes where your data can be found and is in final format, as it will be published as written there

8) I have noticed in your acknowledgements that you thank Iolanda Domingues for input on flow cytometry. However, on the study I did not find a mention of its use. If I am mistaken, we would require the FCS files.

9) Per journal policy, if you have generated any custom code during the course of this investigation, please make it available without restrictions. Please ensure that the code is sufficiently well documented and reusable, and that your Data Statement in the Editorial Manager submission system accurately describes where your code can be found. More information on our Code Policy, what and how to share can be found here: https://journals.plos.org/plosbiology/s/code-availability

We expect to receive your revised manuscript within two weeks.

*Published Peer Review History*

*Press*

Sincerely,

Melissa

Melissa Vazquez Hernandez, Ph.D.

Associate Editor

PLOS Biology

Reviewer #1:

The authors have put in considerable efforts and fully resolved my last remaining critique. I want to congratulate them on a beautiful manuscript.

---

## [Editor Report · Decision Letter 4]

2 Feb 2026

Dear David,

Thank you for the submission of your revised Research Article "CRISPR-Cas is beneficial in plasmid competition, but limited by competitor toxin-antitoxin activity when horizontally transferred." for publication in PLOS Biology. On behalf of my colleagues and the Academic Editor, Tobias Bollenbach, I am pleased to say that we can in principle accept your manuscript for publication, provided you address any remaining formatting and reporting issues. These will be detailed in an email you should receive within 2-3 business days from our colleagues in the journal operations team; no action is required from you until then. Please note that we will not be able to formally accept your manuscript and schedule it for publication until you have completed any requested changes.

PRESS

Sincerely,

Melissa

Melissa Vazquez Hernandez, Ph.D., Ph.D.

Associate Editor

PLOS Biology
